# Psychosocial Predictors of COVID-19 Vaccine Uptake among Pregnant Women: A Cross-Sectional Study in Greece

**DOI:** 10.3390/vaccines11020269

**Published:** 2023-01-26

**Authors:** Petros Galanis, Irene Vraka, Aglaia Katsiroumpa, Olga Siskou, Olympia Konstantakopoulou, Eleftheria Zogaki, Daphne Kaitelidou

**Affiliations:** 1Clinical Epidemiology Laboratory, Faculty of Nursing, National and Kapodistrian University of Athens, 11527 Athens, Greece; 2Department of Radiology, P. & A. Kyriakou Children’s Hospital, 11527 Athens, Greece; 3Department of Tourism Studies, University of Piraeus, 18534 Piraeus, Greece; 4Center for Health Services Management and Evaluation, Faculty of Nursing, National and Kapodistrian University of Athens, 11527 Athens, Greece; 5Faculty of Midwifery, University of West Attica, West Attica, 12243 Aigaleo, Greece

**Keywords:** COVID-19, pregnant women, vaccine, uptake, predictors

## Abstract

An understanding of the factors associated with the COVID-19 vaccine uptake in pregnant women is paramount to persuade women to get vaccinated against COVID-19. We estimated the vaccination rate of pregnant women against COVID-19 and evaluated psychosocial factors associated with vaccine uptake among them. We conducted a cross-sectional study with a convenience sample. In particular, we investigated socio-demographic data of pregnant women (e.g., age, marital status, and educational level), COVID-19 related variables (e.g., previous COVID-19 diagnosis and worry about the side effects of COVID-19 vaccines), and stress due to COVID-19 (e.g., danger and contamination fears, fears about economic consequences, xenophobia, compulsive checking and reassurance seeking, and traumatic stress symptoms about COVID-19) as possible predictors of COVID-19 vaccine uptake. Among pregnant women, 58.6% had received a COVID-19 vaccine. The most important reasons that pregnant women were not vaccinated were doubts about the safety and effectiveness of the COVID-19 vaccines (31.4%), fear that COVID-19 vaccines could be harmful to the fetus (29.4%), and fear of adverse side effects of COVID-19 vaccines (29.4%). Increased danger and contamination fears, increased fears about economic consequences, and higher levels of trust in COVID-19 vaccines were related with vaccine uptake. On the other hand, increased compulsive checking and reassurance seeking and increased worry about the adverse side effects of COVID-19 vaccines reduced the likelihood of pregnant women being vaccinated. An understanding of the psychosocial factors associated with increased COVID-19 vaccine uptake in pregnant women could be helpful for policy makers and healthcare professionals in their efforts to persuade women to get vaccinated against COVID-19. There is a need for targeted educational campaigns to increase knowledge about COVID-19 vaccines and reduce vaccine hesitancy in pregnancy.

## 1. Introduction

In the 21st century, several issues threaten individuals’ health and quality of life and may have an impact on habit change, attitudes, and mental health. Among others, climate changes, natural resources depletion, armed conflicts, and the COVID-19 pandemic are emerging priorities in the frame of the 21st century that could affect peoples’ decisions on several issues including vaccination [1]. In particular, climate changes and natural resource depletion are affecting our environment, healthcare, and economies [2,3,4], causing physical and mental health problems, e.g., anxiety, depression, fear, and stress sleep disturbances [5,6,7,8]. Moreover, at the start of 2023, 46 armed conflicts are counted worldwide. Of these, 19 show high or extreme levels of conflict severity [9]. The literature suggests the negative relationship between war or/and armed conflicts and mental health since the populations involved experience a higher risk of post-traumatic stress disorder, anxiety, depression, substance use, stress, loneliness, and other related conditions [10,11,12,13]. Additionally, in the last three years, the world has been facing the threat of the COVID-19 pandemic, with more than 6.7 million deaths and more than 670 million cases caused by the disease as of 23 January 2023 [14]. Several systematic reviews confirm the negative impact of the COVID-19 pandemic on individuals’ health since the mental health consequences of the pandemic could be comparable to armed conflicts and major disasters [15,16,17].

Unvaccinated pregnant women with symptomatic COVID-19 are at increased risk for iatrogenic preterm births, intensive care unit admission, and invasive ventilation [18,19,20]. In addition, risk of death for unvaccinated women with symptomatic COVID-19 during pregnancy is higher than non-pregnant women with symptomatic COVID-19 [21]. Moreover, unvaccinated pregnant women have been found to have a higher risk of hospital admission for COVID-19 than vaccinated pregnant women [22].

Pregnant women were excluded from early randomized controlled trials, leading to the lack of safety and efficacy data [23,24]. However, the most recent data are rather encouraging for vaccinated pregnant women. In particular, COVID-19 vaccine research reveals that (a) abortion rate, adverse pregnancy, and adverse neonatal outcomes are similar in vaccinated and non-vaccinated pregnant women, (b) anti-SARS-CoV-2 immunoglobulins provide immunity to the newborns, and (c) COVID-19 vaccines do not cause vaccine-related adverse events [25,26,27,28,29]. Thus, several organizations worldwide now recommend COVID-19 vaccination for pregnant women and women who are trying to become pregnant, or who might become pregnant in the future to prevent severe maternal morbidity and adverse birth outcomes [30,31].

Until now, several studies investigated the determinants of COVID-19 vaccine uptake among pregnant women [29,32,33,34,35,36,37,38,39,40,41,42,43,44,45]. So far, research has focused on the impact of the socio-demographic characteristics on the decision of pregnant women to be vaccinated against COVID-19, e.g., age, ethnicity, race, education, and income. Moreover, emphasis is given on COVID-19-related variables and beliefs towards vaccination, such as COVID-19 infection during pregnancy, receipt of sufficient evidence and advice, recommendation by public health officials and healthcare workers, trust in vaccines and the health system, vaccine hesitancy, and previous refusal of the influenza vaccine. However, only a few studies focused on the psychosocial predictors of COVID-19 vaccine uptake among pregnant women, e.g., trust in COVID-19 vaccines, fear of COVID-19, and worry about the adverse side effects of the COVID-19 vaccines [39,45,46].

Thus, the aim of our study was to estimate the rate of vaccinated pregnant women against COVID-19 and to study the psychosocial predictors of COVID-19 vaccine uptake. In particular, we estimated the COVID-19 vaccination uptake in a sample of pregnant women in Greece one year after the onset of vaccination against COVID-19 in the country. Moreover, we investigated the following research hypotheses: (a) socio-demographic data of pregnant women (e.g., age, marital status, and educational level) could have an impact on vaccination uptake, (b) COVID-19 related variables (e.g., previous COVID-19 diagnosis, and worry about the side effects of COVID-19 vaccines) could affect pregnant women’s decisions to receive a COVID-19 vaccine, and (c) COVID-19-related stress (e.g., danger and contamination fears, fears about economic consequences, and traumatic stress symptoms about COVID-19) could be predictor of vaccination uptake.

## 2. Materials and Methods

### 2.1. Study Design and Participants

We conducted a cross-sectional study with a convenience sample in Greece. Recruitment of pregnant women began in December 2021, while the Greek government had offered a free COVID-19 vaccine to all pregnant women from April 2021 [47]. In particular, we collected data from December 2021 to March 2022. We used several approaches to collect our data. Firstly, we used Google forms to create an anonymous version of the study questionnaire and then we disseminated it through social media, i.e., Facebook, WhatsApp, and LinkedIn. Moreover, we sent the questionnaire via e-mail to all our contacts, searching for pregnant women. Our message emphasized that only pregnant women could participate in our study, excluding all other population groups. At the same time, we asked pregnant women who completed the study questionnaire to invite to our study other pregnant women they might have known. Thus, a snowball sampling method was applied. Women were considered to be eligible for the study if they were pregnant, aged 18 or older, and indicated that they could complete the study questionnaire in Greek. Data were collected anonymously and kept confidential and participation in the survey was voluntary. Pregnant women were informed that they could cease to participate at any stage of the study should they wished to do so.

During the study period, four COVID-19 vaccines (Pfizer/BiotNTech, Moderna, Vaxzevria/AstraZeneca, and Janssen/Johnson & Johnson) were offered free of charge for all citizens in Greece. Since data from pregnant women for Vaxzevria/AstraZeneca and Janssen/Johnson & Johnson were limited until data collection time, only Pfizer/BiotNTech and Moderna vaccines were offered by the Greek Ministry of Health for pregnant women. Pregnant women at any number of pregnancy week could take the primary COVID-19 vaccine doses according to the clinical guidelines. Moreover, conceiving prior to completion of COVID-19 vaccination did not affect the vaccination schedule. Mixing types of COVID-19 vaccines was not allowed. Changes in dosing interval were also allowed, e.g., in case of a SARS-CoV-2 infection. An updated version of these clinical guidelines was released on November 2022 from the Greek Ministry of Health [47]. The COVID-19 vaccination was not compulsory in order to deliver at public healthcare facilities. In addition, COVID-19 vaccination did not have any implications on medical insurance during delivery.

Considering a low effect size (odds ratio = 1.4), the precision level as 5% (alpha level), and the power as 95%, a minimum sample size of 564 pregnant women was required. Moreover, 666 pregnant women were required in order to achieve a vaccination uptake rate of 50%, a confidence level of 99%, and a margin of error of 5% since the reference population of fully vaccinated people in Greece was 7.6 million. Nevertheless, we strove to recruit a larger number of pregnant women to further decrease random error. 

The Ethics Committee of Department of Nursing, National and Kapodistrian University of Athens approved the study protocol (reference number; 370, 2 September 2021).

### 2.2. Questionnaires

Firstly, we collected the following socio-demographic data of pregnant women: age (continuous), marital status (singles, married, in a couple relationship without marriage, or divorced), educational level (elementary school, high school, or university degree), self-perceived financial status (very poor, poor, average, good, or very good), self-perceived physical health (very poor, poor, average, good, or very good), having children aged 18 or younger (no or yes), chronic disease (no or yes), previous COVID-19 diagnosis (no or yes), and family members/friends with previous COVID-19 diagnosis (no or yes).

In addition, we used four questions to measure pregnant women’s vaccination status and COVID-19-related vaccination status. In particular, we asked pregnant women (1) if they had received vaccination against the seasonal influenza (no or yes), (2) if they had received vaccination against the COVID-19 (no or yes), and (3) if they had received vaccination against the COVID-19 before or during pregnancy (no or yes). Moreover, we asked unvaccinated pregnant women to state the main reason for this denial (doubts about the safety and effectiveness of COVID-19 vaccines, fear of adverse side effects of COVID-19 vaccines, low self-perceived threat regarding COVID-19, previous COVID-19 diagnosis, fear due to a chronic disease, family doctor’s recommendation due to physical health of pregnant women, religious issues, or fear that the COVID-19 vaccination could be harmful to the fetus). 

We measured pregnant women’s worry about the side effects of COVID-19 vaccines with a single item: “I am worried about the side effects that COVID-19 vaccines can have”. Also, we measured pregnant women’s trust in COVID-19 vaccines with a single item: “I trust the COVID-19 vaccines”. Answers in both cases were indicated on a scale ranging from 0 (totally disagree) to 10 (totally agree).

We used the 36-item COVID-19 Stress Scales (CSS) to measure COVID-19-related stress [48]. In particular, we used the Greek version of the CSS [49] since the Greek language is the official language of Greece, spoken by the majority of the population. The questionnaire consists of five factors: (1) danger and contamination fears, (2) fears about economic consequences, (3) xenophobia, (4) compulsive checking and reassurance seeking, and (5) traumatic stress symptoms about COVID-19. Responses for the first three factors are indicated on a 5-point Likert scale ranging from 0 (not at all) to 4 (extremely), and for the last two factors on a 5-point Likert scale ranging from 0 (never) to 4 (almost always). For each factor, a stress score was calculated by averaging the answers to all items of the factor. Thus, score for each factor ranges from 0 to 4 with higher scores reflecting greater stress. We performed a validation analysis for the CSS. In particular, we calculated Cronbach’s alpha coefficient in order to estimate the reliability of the questionnaire. Additionally, we performed transcultural validation and confirmatory factor analysis to investigate the validity of the CSS. We applied the forward–backward–translation method to achieve the transcultural validation of the CSS [50]. In particular, one language expert translated the CSS from English to Greek (forward-translation) and then another language expert translated the tool from Greek back to English. A third senior scholar checked for discrepancies. We used AMOS (version 23) to perform confirmatory factor analysis. The literature suggests the following cut-off values as good fit indices: chi-square divided by degree of freedom (x^2^/df) < 5; root mean square error of approximation (RMSEA) < 0.08; goodness-of-fit index (GFI), adjusted goodness-of-fit index (AGFI), Tucker-Lewis index (TLI), incremental fit index (IFI), normed fit index (NFI), and comparative fit index (CFI) > 0.90 [51,52,53,54]. 

Moreover, all parts of the study questionnaire were tested for face validity. First, we performed interviews with 10 pregnant women in order to perform cognitive testing of the questionnaire [55,56]. All participants understood instructions, questions, and answers. In addition, prior to the final study we conducted a pilot study with 25 pregnant women in order to confirm the face validity of the study questionnaire. Again, we did not uncover issues related to comprehension and cultural relevance.

### 2.3. Statistical Analysis

We used numbers and percentages to present categorical variables and mean and standard deviation to present continuous variables. Socio-demographic data of pregnant women, COVID-19-related stress, worry about the adverse side effects of the COVID-19 vaccines, and trust in the COVID-19 vaccines were the independent variables in our study. The outcome variable was COVID-19 vaccination uptake among pregnant women, measured through “yes/no” answers. Thus, we used logistic regression analysis to identify the predictors of COVID-19 vaccine uptake. We performed univariate and multivariable logistic regression analysis, calculating unadjusted and adjusted odds ratios, respectively. We also calculated 95% confidence intervals (CI) and two-sided *p*-values. We applied the backward elimination model to create the final multivariable logistic regression model. We used independent sample *t*-tests to compare vaccinated and unvaccinated groups regarding the level of danger and contamination fears, fears about economic consequences, xenophobia, compulsive checking and reassurance seeking, and traumatic stress symptoms about COVID-19. A *p*-value < 0.05 was considered as statistically significant. Statistical analysis was performed using the Statistical Package for Social Sciences software (IBM Corp. Released 2012. IBM SPSS Statistics for Windows, Version 21.0. Armonk, NY, USA, IBM Corp.).

## 3. Results

### 3.1. Socio-Demographic Characteristics

The study population included 812 pregnant women. The pregnant women’s mean age was 31.6 years, while 35.3% had children less than 18 years old. The majority of the participants were married or in a couple relationship without marriage (89.7%). Among the pregnant women, 48.3% defined their financial status as good/very good. Almost 15% of the pregnant women suffered from a chronic disease and 90.5% defined their health status as good/very good. In total, 24.1% of the pregnant women and 79.3% of their family members/friends were diagnosed with COVID-19. Detailed socio-demographic characteristics of the pregnant women are shown in Table 1.

### 3.2. COVID-19-Related Vaccination Status

Among the pregnant women in our study, 58.6% had received a COVID-19 vaccine. Among the vaccinated women, 45.6% had received a COVID-19 vaccine during pregnancy. The most important reasons that pregnant women were not vaccinated were doubts about the safety and effectiveness of COVID-19 vaccines (31.4%), fear that COVID-19 vaccines could be harmful to their fetus (29.4%), and fear of adverse side effects of COVID-19 vaccines (29.4%). In our sample, 24.1% of the pregnant women had received a flu vaccine during 2021. COVID-19-related vaccination status of the pregnant women is presented in Table 2.

### 3.3. Validation of the COVID-19 Stress Scales

Reliability of the CSS in our study was excellent. In particular, Cronbach’s alpha coefficients for the “danger and contamination fears”, “fears about economic consequences”, “xenophobia”, “compulsive checking and reassurance seeking”, and “traumatic stress symptoms about COVID-19” factors were 0.93, 0.94, 0.95, 0.88, and 0.93, respectively.

Additionally, confirmatory factor analysis confirmed the five factor structure of the CSS, since all fitting indices were good: x^2^/df = 3.665; RMSEA = 0.057; GFI = 0.933; AGFI = 0.908; TLI = 0.949; IFI = 0.975; NFI = 0.966; CFI = 0.975 (Figure 1).

### 3.4. COVID-19-Related Stress

The highest levels of COVID-19-related stress were due to danger and contamination fears and then to xenophobia, compulsive checking and reassurance seeking, fears about economic consequences, and traumatic stress symptoms about COVID-19 was low. The level of danger and contamination fears was moderate for both vaccinated and unvaccinated pregnant women, while the level of (1) fears about economic consequences, (2) xenophobia, (3) compulsive checking and reassurance seeking, and (4) traumatic stress symptoms about COVID-19 was low. The mean level of danger and contamination fears (*p* < 0.001), fears about economic consequences (*p* = 0.042), xenophobia (*p* = 0.471), compulsive checking and reassurance seeking (*p* = 0.401), and traumatic stress symptoms about COVID-19 (*p* = 0.384) among the vaccinated pregnant women was higher than that of the unvaccinated pregnant women. Differences regarding danger and contamination fears and fears about economic consequences were statistically significant. COVID-19-related stress of pregnant women according to COVID-19 vaccination status is shown in Table 3. 

### 3.5. Predictors of COVID-19 Vaccine Uptake

Multivariable logistic regression analysis identified that five psychosocial factors and seven socio-demographic characteristics affect a pregnant woman’s decision to receive a COVID-19 vaccine. The multivariable logistic regression analysis for COVID-19 vaccine uptake among the pregnant women revealed that the independent variables explained 55.6% of the variance in this outcome. Predictors of the COVID-19 vaccine uptake are shown in Table 4. 

Regarding psychosocial factors, increased danger and contamination fears, increased fears about economic consequences, and higher levels of trust in COVID-19 vaccines were related with pregnant women’s COVID-19 vaccine uptake. On the other hand, increased compulsive checking and reassurance seeking and increased worry about the adverse side effects of COVID-19 vaccines reduced the likelihood of pregnant women being vaccinated against COVID-19. 

Regarding socio-demographic characteristics, single women, women who already had at least one child aged <18 years old, women who defined their health status as very poor/poor/moderate, women without a previous COVID-19 diagnosis, women with family members/friends with previous COVID-19 diagnosis, and women who were vaccinated against the seasonal influenza had a greater probability to be vaccinated against COVID-19. Moreover, increased age was significantly associated with higher COVID-19 vaccine uptake among pregnant women.

## 4. Discussion

To the best of our knowledge, this is the first study that investigates psychosocial predictors of COVID-19 vaccine uptake among pregnant women with a valid questionnaire in Greek. Three other studies investigated a few psychosocial predictors but without the use of a valid questionnaire [39,45,46]. In a sample of pregnant women in Greece, we found that 58.6% of them were vaccinated against COVID-19 during pregnancy. Papazachariou et al. (2023) found that 50.6% of females had been vaccinated for COVID-19 [57]. A recent meta-analysis found that the overall proportion of vaccinated pregnant women against COVID-19 was 27.5% [58]. Moreover, it is interesting to highlight the fact that the vaccination rate is much higher in Israel (43.3%) than in the USA (27.3%) and other countries (12.8%) [58]. Additionally, several other studies found a lower vaccination rate among pregnant women in Italy (49.4%) [44], Spain (56%) [45], Wales (32.7%) [59], Canada (42.4%) [41], and the USA (44%) [60]. The higher vaccination rate in our study may be due to the fact that the studies in the meta-analysis were conducted from December 2020 to September 2021, while our study was conducted from December 2021 to March 2022. Evidence on the safety and efficacy of the COVID-19 vaccines is constantly increasing which may also increase the confidence of pregnant women in vaccines.

We found that the main reasons that pregnant women were not vaccinated were doubts about the safety and effectiveness of COVID-19 vaccines, fear that COVID-19 vaccines could be harmful to their fetus, and fear of adverse side effects. This finding is confirmed by the literature, since several studies have revealed women’s fears that COVID-19 vaccines can cause problems in fertility, pregnancy, and breastfeeding [61,62,63]. Moreover, pregnant women are more worried about lack of safety data in pregnancy, fetal effects, vaccine adverse side effects, and rushed development of vaccines [39,45,46,64]. Moreover, worry in the general population about safety, efficacy, and side effects of COVID-19 vaccines is related to hesitancy in COVID-19 vaccine uptake [65].

According to our multivariate analysis, several psychosocial factors affect a pregnant woman’s decision to receive a COVID-19 vaccine. In particular, increased danger, contamination fears, and fears about economic consequences were related with COVID-19 vaccine uptake among pregnant women. Siegel et al. (2022) [39] confirm our finding, since they found that vaccinated pregnant women are more afraid of COVID-19 during pregnancy than unvaccinated pregnant women. Moreover, vaccinated pregnant women believe that if they are infected, they are at risk for getting very sick [39]. Increased fear of pregnant women in our study may also be due to the fact that they have family members/friends with a previous COVID-19 diagnosis. Siegel et al. (2022) confirm this result since vaccinated pregnant women are more likely to know hospitalized COVID-19 patients [39]. In general, fear of COVID-19 is associated with good prevention practices in both pregnant women and the general population [66,67,68,69,70]. Probably, individuals who have a fear of acquiring COVID-19 and the spread of it could better adopt a protective health behavior againstCOVID-19 and comply with preventive measure recommendations. Therefore, fear appears to promote preventive measures during the COVID-19 pandemic, e.g., frequent hand-washing, physical distancing, face masks, and vaccination.

On the other hand, we found that high levels of COVID-19-related stress and, more specifically, the stress caused by the compulsive checking and reassurance seeking reduce the COVID-19 vaccine uptake among the pregnant women. Pregnant women experience moderate to high levels of stress during the COVID-19 pandemic [71,72,73]. Particular attention should be focused on pregnant women since COVID-19-related stress is associated with depressive symptoms [74]. Higher depressive symptoms in pregnancy may have negative effects on infants’ health, e.g., disruptions of pro- and anti-inflammatory markers, infant stress responses, shortened breastfeeding period, and poorer bonding between mothers and their infants [75,76,77,78,79]. Moreover, there is a negative association between perceived stress and self-care behaviors among pregnant women [67]. Negative self-care behaviors among pregnant women with high levels of COVID-19-related stress could explain the low vaccination rate among those women since self-care behaviors are purposeful actions that individuals take on to improve their health [80,81]. 

Moreover, we found that higher levels of trust in COVID-19 vaccines were associated with higher COVID-19 vaccine uptake among pregnant women. The literature confirms this finding, since Siegel et al. (2022) found that trust in COVID-19 vaccines’ effectiveness for women and newborns is associated with increased vaccine uptake [39]. In addition, they recognized that unvaccinated pregnant women are less likely to trust vaccine developers. Moreover, general distrust is related with hesitancy in COVID-19 vaccine uptake in the general population [65]. It is noteworthy that the mistrust in the COVID-19 vaccines is higher among individuals from ethnic minorities, both in the general population and among pregnant women [82,83,84]. Another important issue is that several studies found a positive relation between trust in COVID-19 vaccines and parents’ willingness to vaccinate their children against COVID-19 [64,85,86,87]. 

Our study identified several socio-demographic predictors of COVID-19 vaccination uptake among pregnant women. Among others, we found that pregnant women who received a flu vaccine had also a greater probability to be vaccinated against COVID-19. Systematic reviews have already shown that previous seasonal influenza vaccination history is a strong predictive factor for individuals to accept a COVID-19 vaccine, both for themselves and their children [85,88,89,90,91]. Moreover, recent studies expand this evidence by confirming that COVID-19 vaccination uptake is more common among individuals who are already vaccinated against the seasonal influenza [92,93,94,95]. Influenza vaccination during the COVID-19 pandemic is crucial since it may have a protective role against the COVID-19 pandemic by reducing the SARS-CoV-2 infection rate, hospitalization, severity of COVID-19, admission to intensive care units, and mortality [96,97].

Older age of pregnant women is another predictor of COVID-19 vaccination uptake. Our findings confirm the evidence that older age is associated with increased likelihood of acceptance and uptake of a COVID-19 vaccine among pregnant women [38,40,64,98]. Probably, older pregnant women experience more fear of COVID-19 since the evidence shows that pregnancy at advanced maternal age is a strong predictor for adverse outcomes, e.g., neonatal intensive care unit admission, low birth weight babies, worse Apgar scores, spontaneous miscarriage, cesarean deliveries, and pre-eclampsia [99,100]. Moreover, it is well known that older age increases adverse outcomes in COVID-19 patients, such as hospitalization, intensive care unit admission, and mortality [101,102,103]. This fear that pregnant women experience against COVID-19 is confirmed by two more findings in our study. In particular, we found that pregnant women who defined their health status as poor and women without a previous COVID-19 diagnosis were more likely to get vaccinated against the COVID-19. In a similar way, Blakeway et al. (2022) found that pre-gestational diabetes mellitus is associated with COVID-19 vaccination uptake among pregnant women [32]. Higher levels of fear about contracting COVID-19 increase the intention of individuals to accept a COVID-19 vaccine [104,105]. For instance, intention of people to accept a COVID-19 vaccine is higher during the lockdown periods when the self-perceived COVID-19 vulnerability is higher [106,107]. Moreover, past COVID-19 patients are less likely to be vaccinated, since they perceive a low risk and feel protected against COVID-19 [108]. 

### Limitations

We should note a number of limitations in our study. Firstly, we conducted a cross-sectional study and therefore we are unable to establish a causal mechanism between psychosocial factors and COVID-19 vaccination uptake among pregnant women. Second, we relied on a convenience sample of pregnant women who cannot be considered representative of the population of pregnant women in Greece. For instance, the educational level of the participants in our study was high, while the participation rate of migrants was probably low since the questionnaire was only in the Greek language. Indicative, a study with 1700 pregnant women in Athens, Greece found that 25.6% possessed a university degree [109] while the respective percentage in our study was much higher (73.3%). Moreover, in a study in Greece with a nationally representative sample of pregnant women, 46.1% reported a low annual family income, 45.2% reported a moderate level, and 8.7% reported a high level [110]. In contrary, in our study, 4.3% of pregnant women perceived their financial status as very poor/poor, 47.4% as moderate, and 48.3% as good/very good. We should notice that mean age of pregnant women in our study is similar to other studies in Greece [109,110,111]. Third, we used a valid questionnaire to measure psychosocial pattern of pregnant women but our data were based on self-reported measures which may introduce information bias due to tendency of participants to seek social desirability. For example, self-perceived financial status and self-perceived physical health could introduce information bias since they are proxies for the measurements. Further studies could measure these variables in a more valid way, e.g., by measuring annual family income instead of self-perceived financial status. Fourth, there are also other psychosocial factors that could affect a pregnant woman’s decision to receive a COVID-19 vaccine, e.g., anxiety, depression, and quality of life. Fifth, we disseminated the study questionnaire through social media and our e-mail contacts. Theoretically, each pregnant woman in Greece could participate in our study given that she had internet access. Although we did not limit our study in geographical terms, it is probable that our participants were mainly living in cities. We hypothesize that participation rate from countryside and islands was low. Sixth, our study population includes only Greek pregnant women since our aim was to evaluate attitudes of natives and not of migrants. Migrants are a group with different attitudes, cultural background, and religion beliefs, and thus, a different study design should be implemented to assess their COVID-19 vaccination uptake. Future research should investigate migrants’ attitudes towards COVID-19 vaccines in order to make a valid comparison with natives. Finally, we did not measure some possible confounders, such as ethnicity, gestational week, at-risk pregnancy, vaccine type (Pfizer/BiotNTech and Moderna) that was offered to pregnant women, number of people in the household, employment status (housewife or employed), and work location (working in person or working remotely). Moreover, several emerging priorities in the 21st century such as climate changes, natural resource depletion, and armed conflicts could affect people’s decisions to accept a COVID-19 vaccine. Thus, the role of these variables should be investigated in future studies in order to obtain more valid results.

## 5. Conclusions

Our study is the first to assess psychosocial predictors of COVID-19 vaccine uptake among pregnant women with a valid instrument in Greek. This study is very timely due to the ongoing high COVID-19 case rates globally and the known increased risks of COVID-19 in pregnant women. An understanding of the psychosocial factors associated with increased COVID-19 vaccine uptake in pregnant women could be helpful for policy makers and healthcare professionals in their efforts to persuade women to get vaccinated against COVID-19. Furthermore, we found that several socio-demographic characteristics of pregnant women affect their decision to receive a COVID-19 vaccine. Our findings emphasize the need for a more sensitive approach in the attempt to encourage vaccination uptake in pregnancy, especially in younger women, women with a good self-perceived health status, women with a previous COVID-19 diagnosis, and women who are not vaccinated against the seasonal influenza. Thus, public educational campaigns to encourage COVID-19 vaccination in the population of pregnant women should be targeted, taking into account different concerns, needs, and attitudes. Mass vaccination of pregnant women is paramount to reduce COVID-19 vaccine hesitancy in pregnancy, halt the spread of SARS-CoV-2, and reduce adverse outcomes in mothers and newborns.

Our findings could be an alarm signal for scholars worldwide to investigate the possible role of psychosocial factors on a pregnant woman’s decision to accept a COVID-19 vaccine. Until now, studies have mainly focused on demographic variables as possible predictors of vaccination uptake among pregnant women. Research should be expanded in order to include a variety of determinants of vaccine hesitancy. Modifiable factors, such as knowledge, trust, and information, should be of particular interest. Moreover, we should investigate these issues in vulnerable groups, such as pregnant women in ethnic minority populations, developing countries, migrant populations, and areas of high deprivation. Since vaccine hesitancy is already recognized by the World Health Organization as one of the 10 biggest threads to global health [112], identification of factors that enhance individuals’ intention to accept vaccination is crucial to increase vaccine uptake. Several systematic reviews suggest that vaccination against a variety of diseases (i.e., pneumococcal disease, varicella, tuberculosis) is a cost-effective or cost-saving intervention [113,114,115,116]. Additionally, a recent systematic review showed that the COVID-19 vaccination program appears to be a cost-effective or cost-saving intervention [117]. Therefore, the implementation of effective vaccination programs worldwide is essential to save lives and resources, and improve quality of life.

## Figures and Tables

**Figure 1 vaccines-11-00269-f001:**
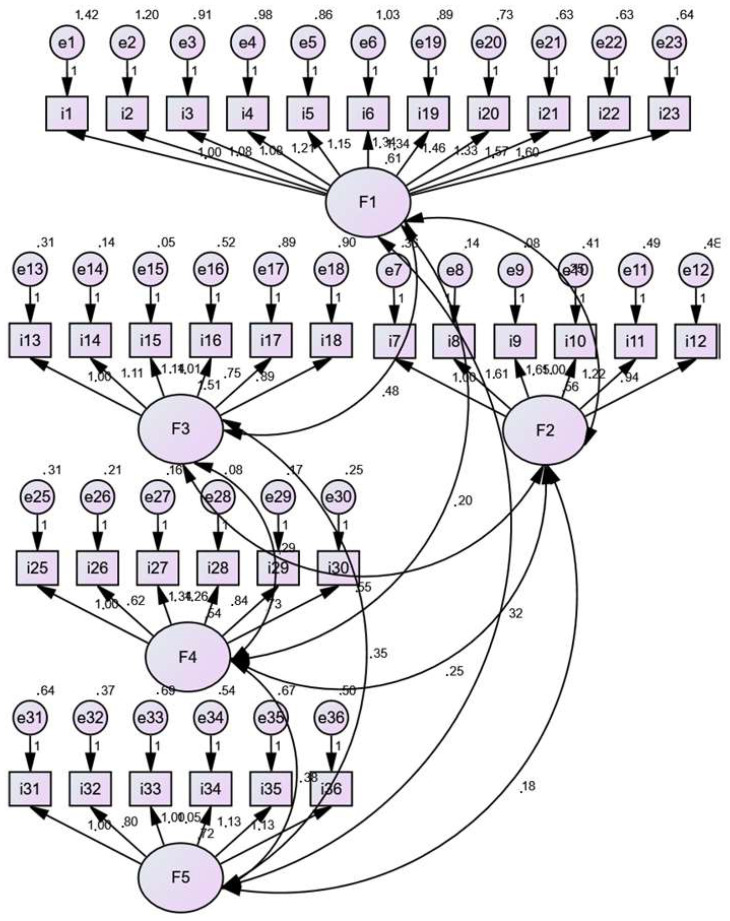
Confirmatory factor analysis for the questionnaire “COVID-19 Stress Scales”.

**Table 1 vaccines-11-00269-t001:** Socio-demographic characteristics of the pregnant women (N = 812).

Variable	N	%
Marital status		
Singles	84	10.3
Married or in a couple relationship without marriage	728	89.7
Divorced	0	0
Age (years), mean, standard deviation	31.6	4.6
Children < 18 years old		
No	525	64.7
Yes	287	35.3
Educational level		
Elementary school	7	0.9
High school	210	25.9
University degree	595	73.3
Self-perceived financial status		
Very poor	7	0.9
Poor	28	3.4
Moderate	385	47.4
Good	336	41.4
Very good	56	6.9
Self-perceived health status		
Very poor	0	0
Poor	0	0
Moderate	77	9.5
Good	462	56.9
Very good	273	33.6
Chronic disease		
No	693	85.3
Yes	119	14.7
Previous COVID-19 diagnosis		
No	616	75.9
Yes	196	24.1
Family members/friends with previous COVID-19 diagnosis		
No	168	20.7
Yes	644	79.3

**Table 2 vaccines-11-00269-t002:** COVID-19-related vaccination status in the pregnant women.

Variable	N	%
COVID-19 vaccination uptake		
No	336	41.4
Yes	476	58.6
COVID-19 vaccination uptake before pregnancy		
No	217	45.6
Yes	259	54.4
Seasonal influenza vaccination in 2021		
No	616	75.9
Yes	196	24.1
Reasons for decline of pregnant women to receive a COVID-19 vaccine		
I have doubts about the safety and effectiveness of COVID-19 vaccines	112	31.4
I am afraid of adverse side effects of COVID-19 vaccines	105	29.4
I believe that even I get infected with COVID-19, nothing bad will happen to me	14	3.9
I have already been diagnosed with COVID-19 and the vaccine will not be beneficial for me	14	3.9
I am afraid because I suffer from a chronic disease	7	2.0
I am afraid that COVID-19 vaccines could be harmful to my fetus	105	29.4

**Table 3 vaccines-11-00269-t003:** COVID-19-related stress of pregnant women according to COVID-19 vaccination status.

Scale	Mean	Standard Deviation	*p*-Value ^a^
Danger and contamination fears			<0.001
Unvaccinated	2.01	1.06	
Vaccinated	2.34	1.02	
Fears about economic consequences			0.042
Unvaccinated	0.51	0.82	
Vaccinated	0.65	1.05	
Xenophobia			0.471
Unvaccinated	0.97	1.24	
Vaccinated	1.04	1.27	
Compulsive checking and reassurance seeking			0.401
Unvaccinated	0.87	0.84	
Vaccinated	0.92	0.97	
Traumatic stress symptoms about COVID-19			0.384
Unvaccinated	0.40	0.74	
Vaccinated	0.45	0.74	

^a^ independent samples *t*-test.

**Table 4 vaccines-11-00269-t004:** Univariate and multivariable logistic regression analysis with COVID-19 vaccine uptake among the pregnant women as the dependent variable (reference: COVID-19 vaccine denial).

Variable	Unadjusted OR (95% CI)	*p*-Value	Adjusted OR (95% CI) ^a^	*p*-Value ^b^
Marital status (singles vs. married)	1.45 (0.91–2.36)	0.12	2.18 (1.01–4.70)	0.046
Age (years)	1.07 (1.04–1.10)	<0.001	1.09 (1.04–1.14)	<0.001
Children aged <18 years old (yes vs. no)	1.36 (1.01–1.83)	0.04	1.88 (1.23–2.89)	0.004
Educational level (University degree vs. high school)	2.75 (1.99–3.79)	<0.001		NS
Self-perceived financial status (very good/good vs. moderate/poor/very poor)	1.18 (0.89–1.56)	0.24		NS
Self-perceived health status (very poor/poor/moderate vs. good/very good)	1.26 (0.78–2.05)	0.35	3.44 (1.69–7.00)	0.001
Chronic disease (yes vs. no)	1.35 (0.90–2.03)	0.15		NS
Previous COVID-19 diagnosis (no vs. yes)	1.31 (0.95–1.81)	0.10	1.63 (1.01–2.64)	0.047
Family members/friends with previous COVID-19 diagnosis (yes vs. no)	2.39 (1.69–3.38)	<0.001	1.79 (1.08–2.96)	0.025
Seasonal influenza vaccination in 2021 (yes vs. no)	8.72 (5.40–14.09)	<0.001	17.64 (8.63–36.17)	<0.001
Danger and contamination fears	1.35 (1.18–1.54)	<0.001	1.43 (1.13–1.79)	0.003
Fears about economic consequences	1.16 (0.99–1.35)	0.05	1.58 (1.24–2.03)	<0.001
Xenophobia	1.04 (0.93–1.17)	0.47		NS
Compulsive checking and reassurance seeking	1.07 (0.91–1.24)	0.41	0.50 (0.39–0.64)	<0.001
Traumatic stress symptoms about COVID-19	1.09 (0.89–1.32)	0.38		NS
Worry about the side effects of COVID-19 vaccines	0.84 (0.81–0.88)	<0.001	0.73 (0.68–0.79)	<0.001
Trust in COVID-19 vaccines	1.36 (1.29–1.44)	<0.001	1.53 (1.41–1.65)	<0.001

An odds ratio <1 indicates a negative association, while an odds ratio >1 indicates a positive association. CI: confidence interval; NS: non-significant; OR: odds ratio ^a^ R^2^ for the final multivariable model was 55.6% ^b^ Statistically significant independent variables after the backward elimination regression analysis.

## Data Availability

The data presented in this study are available on request from the corresponding author.

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
