# Peer review of "Psychosocial Predictors of COVID-19 Vaccine Uptake among Pregnant Women: A Cross-Sectional Study in Greece"

_vaccines, 2023, doi:10.3390/vaccines11020269_

Round 1
Reviewer 1 Report
The manuscript nicely sheds light on Psychosocial predictors of COVID-19 vaccine uptake among pregnant women. The manuscript itself is well organized, but there are some major points and suggestions for what could be done to improve the article.
The authors have tried to justify the work, but the introduction still needs to be enhanced and enriched by more recent references supporting the aim of the study.
- Line 51: check reference [15–19,19–23]
- Table 3 : where is the significance?
-How did you validate your questionnaire?
- It's better to perform Backward Elimination regression.
- English narrative needs to be firmly corrected, please check spelling, grammar errors, and wordiness across the manuscript. Typos, and grammatical mistakes need fixing.
Author Response
Dear Reviewer, we are grateful for your comments. You really help us to improve our manuscript. We apply all your suggestions to our manuscript.
Please, see the responses below and the revised form of our manuscript.
Dear Reviewer, thank you for giving us the opportunity to revise our manuscript entitled "Psychosocial predictors of COVID-19 vaccine uptake among pregnant women: a cross-sectional study in Greece". We would also like to thank you for your insightful comments and suggestions on how to improve our manuscript. We respectfully tried to address the issues raised and to revise our manuscript accordingly. We hope that our revision will reach the high standards of the journal “Vaccines”.
We are grateful for your comments. You really help us to improve our manuscript. We apply all your suggestions in our manuscript.
Also, we made changes in the manuscript according to the other Reviewers’ instructions.
We look forward to hearing from you
Best Regards
The authors
Comments from Reviewer
The manuscript nicely sheds light on Psychosocial predictors of COVID-19 vaccine uptake among pregnant women. The manuscript itself is well organized, but there are some major points and suggestions for what could be done to improve the article.
The authors have tried to justify the work, but the introduction still needs to be enhanced and enriched by more recent references supporting the aim of the study.
Answer: Done
Dear Reviewer, we expand the Introduction section adding more recent studies and information in order to support the aim of our study.
Please, see the third paragraph in the Introduction section.
- Line 51: check reference [15–19,19–23]
Answer: Done
Dear Reviewer, thank you for your sharp eye. We fix our error.
- Table 3 : where is the significance?
Answer: Done
We add p-values. Also, we comment results in section 3.3. COVID-19-related stress. Please, see Table 3.
-How did you validate your questionnaire?
Answer: Done
Dear Reviewer, we performed confirmatory factor analysis in order to investigate the structure of the questionnaire. Also, we performed interviews with 10 pregnant women to check the face validity. Moreover, we calculated Cronbach’s alpha in order to estimate the reliability of the questionnaire. We add this information in the manuscript. Please, see:
Section 2.2. Questionnaires (last paragraph)
Section 3.3. Validation of the COVID Stress Scales
Supplementary figure 1
- It's better to perform Backward Elimination regression.
Answer: Done
Dear Reviewer, we perform Backward Elimination regression. Please, see table 4. Also we make changes in the section 2.3. Statistical analysis.
- English narrative needs to be firmly corrected, please check spelling, grammar errors, and wordiness across the manuscript. Typos, and grammatical mistakes need fixing.
Answer: Done
Dear Reviewer, we fix our manuscript. Please, see throughout the manuscript (track changes with blue color).
Reviewer 2 Report
Reviewer’s feedback:
“Psychosocial predictors of COVID-19 vaccine uptake among pregnant women: a cross-sectional study in Greece”
General:
Reference section: please ensure the completeness of your references according to the stipulated journal format
Needs proofreading to make sure the language structure and grammar are adequately used. There are some minor grammar issues throughout
There is room for improvement in the paper because the literature review needs to be extended and have more research findings discussed in it
Specific:
Line 63: please state specific year
Line 64: please add citation/reference
Line 67: please state all types of social media platform used (not just giving example FB)
Line 86-87: what are your definitions for: financial status (very poor, poor, average, good, or very good); and physical health (very poor, poor, average, good, or very good) = how to pick which one “very poor, poor, average, good, or very good”
For method, did you conduct pilot study? Please explain either the answer is yes or no.
Line 107: …..” we used the Greek version of the CSS” – please state reason of using Greek version rather than English?
Line 134: “while 35.3% were expecting their first child” – where you got the data from? I could only saw “children aged 18 or younger (no or yes)”
Line 150: change word ‘my’
Line 203: please add ‘in Greek’ = ….…. with a valid questionnaire ‘in Greek’
Line 208: please add citation/reference
Line 316: please add ‘in Greek’ = ….…. with a valid instrument ‘in Greek’
Since you stated vaccine type as one of your limitations, briefly include in the background:
What types of covid19 vaccines manufactured by which companies were offered to pregnant women;
When were approved by your Ministry of Health to administer different types of vaccine;
During that data collection time: is clinical guideline on covid19 vaccination during pregnancy released – if yes then what is the edition of the updated version?
The conditions of when the vaccines could be injected i.e. any number of pregnancy week/between which weeks of pregnancy allowed/ at last term/ after deliver/etc.?;
Do they must complete dose 1 and 2, or must take dose 3/booster 1?;
Is the covid19 vaccination compulsory in order to deliver at government facilities?;
Any implications on medical insurance during delivery?;
Mixing types of covid19 vaccines and change of their dosing interval?;
Safety and efficacy of COVID-19 vaccines among pregnant women?;
Conceiving prior to completion of covid19 vaccination?
Best wishes
Author Response
Dear Reviewer, we are grateful for your comments. You really help us to improve our manuscript. We apply all your suggestions to our manuscript.
Please, see the resposes below and the revised form of our manuscript.
Dear Reviewer, thank you for giving us the opportunity to revise our manuscript entitled "Psychosocial predictors of COVID-19 vaccine uptake among pregnant women: a cross-sectional study in Greece". We would also like to thank you for your insightful comments and suggestions on how to improve our manuscript. We respectfully tried to address the issues raised and to revise our manuscript accordingly. We hope that our revision will reach the high standards of the journal “Vaccines”.
We are grateful for your comments. You really help us to improve our manuscript. We apply all your suggestions in our manuscript.
Also, we made changes in the manuscript according to the other Reviewers’ instructions.
We look forward to hearing from you
Best Regards
The authors
Comments from Reviewer
General:
Reference section: please ensure the completeness of your references according to the stipulated journal format
Answer: Done
Please, see the references section.
Needs proofreading to make sure the language structure and grammar are adequately used. There are some minor grammar issues throughout
Answer: Done
Dear Reviewer, we fix our manuscript. Please, see throughout the manuscript (track changes with blue color).
There is room for improvement in the paper because the literature review needs to be extended and have more research findings discussed in it
Answer: Done
Dear Reviewer, we expand the literature review in the Introduction section. Also, we expand the Discussion section explaining more our findings. We add 16 more references in our manuscript. Please, see the Introduction and the Discussion section.
Specific:
Line 63: please state specific year
Answer: Done
We add the following text: “Recruitment of pregnant women began in December 2021…”.
Line 64: please add citation/reference
Answer: Done
We add the citation/reference with number [30].
Line 67: please state all types of social media platform used (not just giving example FB)
Answer: Done
We add the following text: “… i.e. Facebook, WhatsApp, and LinkedIn.”
Line 86-87: what are your definitions for: financial status (very poor, poor, average, good, or very good); and physical health (very poor, poor, average, good, or very good) = how to pick which one “very poor, poor, average, good, or very good”
Answer: Done
Dear Reviewer, you are right. We recognize that measurement of financial status and physical health as self-perceived variables introduce bias in our study. Thus, we add this limitation. We add the following text:
Third, we used a valid questionnaire to measure psychosocial pattern of pregnant women but our data were based on self-reported measures which may introduce information bias due to tendency of participants to seek for social desirability. For example, self-perceived financial status and self-perceived physical health could introduce information bias since they are proxies for the measurements. Further studies could measure these variables in a more valid way, e.g. by measuring annual family income instead of self-perceived financial status.
For method, did you conduct pilot study? Please explain either the answer is yes or no.
Answer: Done
Dear Reviewer, thank you for this comment. We conducted a pilot study. We add this information in the section 2.2. Questionnaires. Please, see the last paragraph in the section 2.2. Questionnaires.
Line 107: …..” we used the Greek version of the CSS” – please state reason of using Greek version rather than English?
Answer: Done
Dear Reviewer, we conducted our study in Greece. Thus, our participants could understand the Greek language.
Line 134: “while 35.3% were expecting their first child” – where you got the data from? I could only saw “children aged 18 or younger (no or yes)”
Answer: Done
Dear Reviewer, we apologize for our mistake. You are right. The percentage “35.3%” refers to “children aged 18 or younger (no or yes)”. We fix it.
Line 150: change word ‘my’
Answer: Done
Dear Reviewer, thank you for your sharp eye. We word ‘my’ with the word “their”.
Line 203: please add ‘in Greek’ = ….…. with a valid questionnaire ‘in Greek’
Answer: Done
Line 208: please add citation/reference
Answer: Done
Line 316: please add ‘in Greek’ = ….…. with a valid instrument ‘in Greek’
Answer: Done
Since you stated vaccine type as one of your limitations, briefly include in the background:
What types of covid19 vaccines manufactured by which companies were offered to pregnant women;
Answer: Done
During the study period, four vaccines (Pfizer/BiotNTech, Moderna, Vaxzevria/AstraZeneca, and Janssen/Johnson & Johnson) were offered free of charge for all citizens in Greece. Since data from pregnant women for Vaxzevria/AstraZeneca and Janssen/Johnson & Johnson were limited until data collection time, only Pfizer/BiotNTech and Moderna vaccines were offered by the Greek Ministry of Health for pregnant women.
We add the above information in the section 2.1. Study design and participants.
When were approved by your Ministry of Health to administer different types of vaccine;
Answer: Done
Recruitment of pregnant women began in December 2021, while the Greek government had offered a free COVID-19 vaccine to all pregnant women from April 2021
We add the above information in the section 2.1. Study design and participants.
During that data collection time: is clinical guideline on covid19 vaccination during pregnancy released – if yes then what is the edition of the updated version?
Answer: Done
Since data from pregnant women for Vaxzevria/AstraZeneca and Janssen/Johnson & Johnson were limited until data collection time, only Pfizer/BiotNTech and Moderna vaccines were offered by the Greek Ministry of Health for pregnant women. Update version of these clinical guidelines was released on November 2022 from the Greek Ministry of Health.
We add the above information in the section 2.1. Study design and participants.
The conditions of when the vaccines could be injected i.e. any number of pregnancy week/between which weeks of pregnancy allowed/ at last term/ after deliver/etc.?;
Answer: Done
Pregnant women at any number of pregnancy week could take the primary COVID-19 vaccine doses according to the clinical guidelines.
We add the above information in the section 2.1. Study design and participants.
Do they must complete dose 1 and 2, or must take dose 3/booster 1?;
Answer: Done
Pregnant women at any number of pregnancy week could take a the primary COVID-19 vaccine doses according to the clinical guidelines.
We add the above information in the section 2.1. Study design and participants.
Is the covid19 vaccination compulsory in order to deliver at government facilities?;
Answer: Done
COVID-19 vaccination was not compulsory in order to deliver at public healthcare facilities. Also, COVID-19 vaccination did not have any implications on medical insurance during delivery.
We add the above information in the section 2.1. Study design and participants.
Any implications on medical insurance during delivery?;
Answer: Done
COVID-19 vaccination was not compulsory in order to deliver at public healthcare facilities. Also, COVID-19 vaccination did not have any implications on medical insurance during delivery.
We add the above information in the section 2.1. Study design and participants.
Mixing types of covid19 vaccines and change of their dosing interval?;
Answer: Done
Mixing types of COVID-19 vaccines was not allowed. Also, changes in dosing interval were allowed, e.g. in case of a SARS-CoV-2 infection.
We add the above information in the section 2.1. Study design and participants.
Safety and efficacy of COVID-19 vaccines among pregnant women?;
Answer: Done
Since data from pregnant women for Vaxzevria/AstraZeneca and Janssen/Johnson & Johnson were limited until data collection time, only Pfizer/BiotNTech and Moderna vaccines were offered by the Greek Ministry of Health for pregnant women.
We add the above information in the section 2.1. Study design and participants.
Conceiving prior to completion of covid19 vaccination?
Answer: Done
Conceiving prior to completion of COVID-19 vaccination did not affect the vaccination schedule.
We add the above information in the section 2.1. Study design and participants.
Reviewer 3 Report
I enjoyed reading the manuscript.
Some comments:
- I would like to see the comparison in characteristics of pregnant women sampled in the study and the pregnant women at the nationally representative level. This is because the sample was selected based on the convenience sampling and not sure how representative the sample is.
- Is there any statistics of the COVID19 vaccine uptake among pregnant women in other countries? Is 58.6% low or high compared to other countries?
- Is the vaccine uptake among pregnant women lower than non-pregnant women in Greece? Any difference in the COVID prevalence between pregnant women and non-pregnant women?
Author Response
Dear Reviewer, we are grateful for your comments. You really help us to improve our manuscript. We apply all your suggestions to our manuscript.
Please, see the attached file and the revised form of our manuscript.

Reviewer 4 Report
Interesting study on psychosocial predictors of COVID-19 vaccine hesitancy during pregnancy. As the authors state, this cross-sectional study cannot be representative of the population of pregnant women in Greece. But some more background information would be desirable. Where in Greece do the participants live? In cities, countryside, islands? For the reader informations about the vaccination strategy during pregnancy in Greece during the evaluated time period would also be helpful. Which vaccines were recommended and used for pregnant women?
Author Response

(The authors gave the same response as above.)

Round 2
Reviewer 1 Report
The authors have sufficiently responded to my comments.
Author Response
Dear Reviewer thank you for your time and your positive response.
Reviewer 2 Report
Reviewer’s feedback: R1
“Psychosocial predictors of COVID-19 vaccine uptake among pregnant women: a cross-sectional study in Greece”
Line 124: …..” we used the Greek version of the CSS” – please state reason of using Greek version rather than English? = please state reason(s) (not just the study conducted in Greece) e.g. Greek language is the official language of Greece, spoken by 99% of the population. A number of non-official, minority languages and some Greek dialects are spoken as well.
Were your targeted respondents must be Greek nationality? Please state
Line 90: “Update version of these clinical guidelines was released on November 2022 from the Greek Ministry of Health” – please add citation/reference
Line 80: ….four vaccines – please add= ….four ‘COVID-19’ vaccines
Line 193: please include the figure 1 in the manuscript (not as Supplementary Figure 1).
Thank you

Author Response
Dear Reviewer, thank you for giving us the opportunity to revise our manuscript entitled "Psychosocial predictors of COVID-19 vaccine uptake among pregnant women: a cross-sectional study in Greece". We would also like to thank you for your insightful comments and suggestions on how to improve our manuscript. We respectfully tried to address the issues raised and revise our manuscript accordingly. We hope that our revision will reach the high standards of the journal “Vaccines”.
We are grateful for your comments. You really help us to improve our manuscript. We apply all your suggestions to our manuscript.
We look forward to hearing from you
Best Regards
The authors
Comment
Line 124: …..” we used the Greek version of the CSS” – please state reason of using Greek version rather than English? = please state reason(s) (not just the study conducted in Greece) e.g. Greek language is the official language of Greece, spoken by 99% of the population. A number of non-official, minority languages and some Greek dialects are spoken as well.
Answer: Done.
Dear Reviewer, we add the reasons we used the Greek version of the CSS as you suggested. Please see the manuscript (fourth paragraph in the section 2.2. Questionnaires).
Comment
Were your targeted respondents must be Greek nationality? Please state
Answer: Done.
Dear Reviewer, thank you for this comment. Please see the Limitations section where we state the reasons for our choice to recruit only Greek women. We add the following text.
Sixth, our study population includes only Greek pregnant women since our aim was to evaluate attitudes of natives and not of migrants. Migrants are a group with different attitudes, cultural background, and religion beliefs and thus a different study design should be implemented to assess their COVID-19 vaccination uptake. Future research should investigate migrants’ attitudes towards COVID-19 vaccines in order to make a valid comparison with natives.
Comment
Line 90: “Update version of these clinical guidelines was released on November 2022 from the Greek Ministry of Health” – please add citation/reference
Answer: Done.
We add the reference [30].
Comment
Line 80: ….four vaccines – please add= ….four ‘COVID-19’ vaccines
Answer: Done.
We add it.
Comment
Line 193: please include the figure 1 in the manuscript (not as Supplementary Figure 1).
Answer: Done.
We include the figure 1 in the manuscript.
Round 3
Reviewer 1 Report
The authors have responded to me comments.
Author Response

(The authors gave the same response as above.)
